# Marginalised Gaussian Processes with Nested Sampling

**Fergus Simpson**[*]
Secondmind
Cambridge, UK
fergus@secondmind.ai

**Vidhi Lalchand**[*]
University of Cambridge, UK
vr308@cam.ac.uk

**Carl E. Rasmussen**
University of Cambridge, UK
cer54@cam.ac.uk

## Abstract

Gaussian Process models are a rich distribution over functions with inductive biases controlled by a kernel function. Learning occurs through optimisation of the kernel hyperparameters using the marginal likelihood as the objective. This work proposes nested sampling as a means of marginalising kernel hyperparameters, because it is a technique that is well-suited to exploring complex, multi-modal distributions. We benchmark against Hamiltonian Monte Carlo on time-series and two-dimensional regression tasks, finding that a principled approach to quantifying hyperparameter uncertainty substantially improves the quality of prediction intervals.

## 1 Introduction

Gaussian processes (GPs) represent a powerful non-parametric and probabilistic framework for performing regression and classification. An important step in using GP models is the specification of a covariance function and in turn setting the parameters of the chosen covariance function. Both these steps are jointly referred to as the *model selection* problem [Rasmussen and Williams, 2006]. The parameters of the covariance function are called hyperparameters as the latent function values take the place of parameters in a GP.

The bulk of the GP literature addresses the model selection problem through maximisation of the GP marginal likelihood $p(\mathbf{y}|\theta)$. This approach called ML-II[2] typically involves using gradient based optimisation methods to yield point estimates of hyperparameters. The posterior predictive distribution is then evaluated at these optimised hyperparameter point estimates. Despite its popularity this classical approach to training GPs suffers from two issues: 1) Using point estimates of hyperparameters yield overconfident predictions, by failing to account for hyperparameter uncertainty, and 2) Non-convexity of the marginal likelihood surface can lead to poor estimates located at local minima. Further, the presence of multiple modes can affect the interpretability of kernel hyperparameters. This work proposes a principled treatment of model hyperparameters and assesses its impact on the quality of prediction uncertainty. Marginalising over hyperparameters can also be seen as a robust approach to the model selection question in GPs.

The form of the kernel function influences the geometry of the marginal likelihood surface. For instance, periodic kernels give rise to multiple local minima as functions with different periodicities can be compatible with the data. Expressive kernels which are derived by adding/multiplying together primitive kernels to encode different types of inductive biases typically have many hyperparameters, exacerbating the local minima problem.

---

[*]Equal contribution

[2]where 'ML' stands for marginal likelihood and 'II' denotes the second level of the hierarchy pertaining to hyperparameters

35th Conference on Neural Information Processing Systems (NeurIPS 2021).

The spectral mixture (SM) kernel proposed in Wilson and Adams [2013] is an expressive class of kernels derived from the spectral density reparameterisation of the kernel using *Bochner's Therorem* [Bochner, 1959]. The SM kernel has prior support over all stationary kernels which means it can recover sophisticated structure provided sufficient spectral components are used. Several previous works [Kom Samo and Roberts, 2015, Remes et al., 2017, 2018, Benton et al., 2019] have attempted to further enhance the flexibility of spectral mixture kernels, such as the introduction of a time-dependent spectrum. However, we postulate that the key limitation in the SM kernel's performance lies not in its stationarity or expressivity, but in the optimisation procedure. The form of the SM kernel gives rise to many modes in the marginal likelihood, making optimisation especially challenging. It therefore presents an excellent opportunity to test out nested sampling's capabilities, as it is an inference technique which is highly effective at navigating multimodal distributions. We note that aside from being successfully applied to graphical models by Murray et al. [2006], nested sampling has been largely overlooked in the machine learning literature.

Below we provide a summary of our main contributions:

- Highlight some of the failure modes of ML-II training. We provide insights into the effectiveness of ML-II training in weak and strong data regimes.
- Present a viable set of priors for the hyperparameters of the spectral mixture kernel.
- Propose the relevance of Nested Sampling (NS) as an effective means of sampling from the hyperparameter posterior [see also Faria et al., 2016, Aksulu et al., 2020], particularly in the presence of a multimodal likelihood.
- We demonstrate that in several time series modelling tasks (where we predict the future given past observations), incorporating hyperparameter uncertainty yields superior prediction intervals.

## 2  Background

This section provides a brief account of marginalised Gaussian process regression, followed by an outline of the spectral mixture kernel [Wilson and Adams, 2013] which we shall use in the numerical experiments.

**Marginalised Gaussian Processes.**  Given some input-output pairs $(X, \mathbf{y}) = \{\mathbf{x_i}, y_i\}_{i=1}^N$ where $y_i$ are noisy realizations of latent function values $f_i$ with Gaussian noise, $y_i = f_i + \epsilon_i$, $\epsilon_i \sim \mathcal{N}(0, \sigma_n^2)$, we seek to infer some as-yet unseen values $y^*$. Let $k_\theta(\mathbf{x_i}, \mathbf{x_j})$ denote a positive definite kernel function parameterized with hyperparameters $\theta$. Following the prescription of Lalchand and Rasmussen [2020], the marginalised GP framework is given by,

$$\text{Hyperprior: } \theta \sim p(\theta), \quad \text{GP Prior: } \mathbf{f}|X, \theta \sim \mathcal{N}(\mathbf{0}, K_\theta), \quad \text{Likelihood: } \mathbf{y}|\mathbf{f} \sim \mathcal{N}(\mathbf{f}, \sigma_n^2 \mathbb{I}), \quad (1)$$

where $K_\theta$ denotes the $N \times N$ covariance matrix, $(K_\theta)_{i,j} = k_\theta(\mathbf{x_i}, \mathbf{x_j})$. The predictive distribution for unknown test inputs $X^\star$ integrates over the joint posterior[3],

$$p(\mathbf{f}^\star|\mathbf{y}) = \iint p(\mathbf{f}^\star|\mathbf{f}, \theta)p(\mathbf{f}|\theta, \mathbf{y})p(\theta|\mathbf{y})d\mathbf{f}d\theta \tag{2}$$

For Gaussian noise, the integral over $\mathbf{f}$ can be treated analytically as follows

$$p(\mathbf{f}^\star|\mathbf{y}) = \int p(\mathbf{f}^\star|\mathbf{y}, \theta)p(\theta|\mathbf{y})d\theta \simeq \frac{1}{M}\sum_{j=1}^M p(\mathbf{f}^\star|\mathbf{y}, \theta_\mathbf{j}) = \frac{1}{M}\sum_{j=1}^M \mathcal{N}(\mu_j^\star, \Sigma_j^\star),$$

where $\theta$ is dealt with numerically and $\theta_j$ correspond to draws from the hyperparameter posterior $p(\theta|\mathbf{y})$. The distribution inside the summation represents the standard GP posterior predictive for a fixed hyperparameter setting $\theta_j$. Thus, the final form of the posterior predictive in marginalised GPs is approximated by a mixture of Gaussians (see Rasmussen and Williams [2006] for a review of GPs).

Throughout this work we shall adopt a Gaussian likelihood, hence the only intractable integrand we need to consider is the hyperparameter posterior $p(\theta|\mathbf{y})$,

$$p(\theta|\mathbf{y}) \propto p(\mathbf{y}|\theta)p(\theta) \tag{3}$$

---

[3]Implicitly conditioning over inputs $X, X^\star$ for compactness.

For non-Gaussian likelihoods, one would have to approximate the joint posterior over $\mathbf{f}$ and $\theta$.

In related work, Filippone and Girolami [2014] compared a range of MCMC algorithms for marginalised Gaussian processes. Murray and Adams [2010] used a slice sampling scheme for general likelihoods specifically addressing the coupling between $\mathbf{f}$ and $\theta$. Titsias and Lázaro-Gredilla [2014] presented an approximation scheme using variational inference. Jang et al. [2018] considered a Lévy process over the spectral density. Most recently, Lalchand and Rasmussen [2020] demonstrated marginalisation with HMC and variational inference.

**Spectral Mixture Kernels.** The spectral mixture kernel is of particular interest in this work, due to its complex likelihood surface, thereby offering a difficult challenge for sampling methods and optimisation methods alike. The kernel is characterised by a spectral density $S(\nu)$, defined as a mixture of $Q$ Gaussian pairs [Wilson and Adams, 2013]:

$$S(\nu) = \sum_{i=1}^{Q} \frac{w_i}{2} \left[ G(\nu, \mu_i, \sigma_i) + G(\nu, -\mu_i, \sigma_i) \right] . \tag{4}$$

Here the weight $w_i$ specifies the variance contributed by the $i$th component while $G(\nu, \mu_i, \sigma_i)$ denotes a Gaussian function with mean $\mu_i$ and standard deviation $\sigma_i$. To avoid confusion with other standard deviations, and convey its physical significance, we shall refer to $\sigma_i$ as the *bandwidth*.

The formalism of (4) is readily expanded to higher dimensions. If the power spectrum is represented by a sum of multivariate Gaussians, each with a diagonal covariance, with entries collected in the $D$ dimensional vector $\sigma_i$, the corresponding kernel $k(\tau = \mathbf{x} - \mathbf{x}')$ takes the form,

$$k(\tau) = \sum_{i=1}^{Q} w_i \cos(2\pi\tau \cdot \mu_{\mathbf{i}}) \prod_{d=1}^{D} \exp(-2\pi^2 \tau_d^2 \sigma_i^{2(d)}) , \tag{5}$$

where $\tau_d, \mu_i^{(d)}, \sigma_i^{(d)}$ are the $d^{th}$ elements of the $D$ dimensional vectors $\tau, \mu_i$ and $\sigma_i$ respectively. The vector of kernel hyperparameters $\theta = \{w_i, \mu_i, \sigma_i\}_{i=1}^{Q}$ is typically unknown, we account for this uncertainty by treating them as random variables.

**Sampling and Symmetries.** Fig. 1 shows the marginal likelihood surface for a 2-component SM kernel, given two datasets of different size. One of the striking features of these surfaces lies in their symmetry: this is due to the kernel's invariance to the ordering of its components. The marginal likelihood of a SM kernel with $Q$ spectral components possesses $Q!$ identical regions of parameter space. A naive attempt to explore the full posterior distribution of a spectral mixture kernel would try to quantify probability mass across these degenerate regions, a much more computationally intensive task than is necessary. One solution is to only sample from one region, and ignore its symmetric counterparts. To achieve this, we can adopt an approach known as forced identifiability [Buscicchio et al., 2019] to ensure that the components are defined in sequential order with respect to their frequency $\mu$.

While sampling from degenerate (symmetric) modes will not improve performance, in low data regimes the modes are not always well-identified as the LML surface favours several local optima. Hence, a sampling scheme that is capable of sampling from multiple modes will give rise to a diversity of functions compatible with the data.

## 3 Applying Nested Sampling to Gaussian Processes

This section summarises our proposal for how Nested Sampling (NS) can be used to marginalise the hyperparameters of Gaussian Processes, while simultaneously yielding an estimate of the model evidence.

**Nested Sampling.** The nested sampling algorithm was developed by Skilling [2004] (see also Skilling et al. [2006]) as a means of estimating the model evidence $\mathcal{Z} = \int \psi(\theta)\pi(\theta)d\theta$. Here $\pi(\theta)$ denotes the prior, and $\psi$ is the likelihood, which for our purposes is given by the GP marginal likelihood $p(\mathbf{y}|\theta)$. Irrespective of the dimensionality of $\theta$, the integral can be collapsed to a one-dimensional integral over the unit interval, $\mathcal{Z} = \int_0^1 \psi(X)dX$. Here $X$ is the quantile function

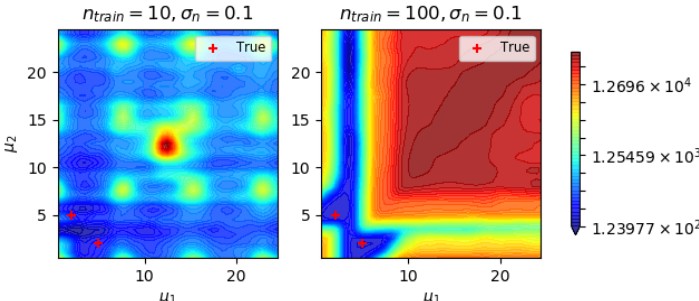

Figure 1: Visualising the negative log marginal likelihood surface as a function of mean frequencies in a 2 component spectral mixture kernel GP. The training data was generated from latent functions evaluated on the input domain [-1,1] and $\sigma_n$ refers to the intrinsic data noise level. The two identical peaks correspond to different re-orderings of the 2 component mean frequency vector with a true value of [2,5].

---

**Algorithm 1:** Nested Sampling for hyperparameter inference

**Initialisation:** Draw $n_L$ 'live' points $\{\boldsymbol{\theta}\}_{i=1}^{n_L}$ from the prior $\boldsymbol{\theta}_i \sim \pi(\boldsymbol{\theta})$, set model evidence $\mathcal{Z} = 0$.
**while** *convergence criterion is unmet* **do**

- Compute $\psi_j = \min(\psi(\boldsymbol{\theta}_1), \dots \psi(\boldsymbol{\theta}_{n_L}))$, the lowest marginal likelihood from the current set of live points.
- Sample a new live point $\boldsymbol{\theta}'$ subject to $\psi(\boldsymbol{\theta}') > \psi_j$
- Remove the point $\boldsymbol{\theta}_i$ corresponding to the lowest marginal likelihood $\psi_j$, moving it to a set of 'saved' points
- Assign estimated prior mass at this step $\hat{X}_j = e^{-\frac{j}{n_L}}$
- Assign a weight for the saved point, $V_j = \hat{X}_{j-1} - \hat{X}_j$
- Accumulate evidence, $\mathcal{Z} = \mathcal{Z} + \psi_j V_j$
- Evaluate convergence criterion, if triggered then break;

**end**
Add final $n_L$ points to the 'saved' list of $K$ samples:

- Each of these final points is assigned a weight $p_i = \hat{X}_K/n_L, \forall i = K, \dots, n_L + K$ // `final slab of enclosed prior mass`
- Final evidence is given by, $\mathcal{Z} = \sum_{i=1}^{n_L+K} \psi_i V_i$
- Importance weights for each sample are given by, $p_i = \psi_i V_i / \mathcal{Z}$

**return** *set of samples* $\{\boldsymbol{\theta}_i\}_{i=1}^{n_L+K}$, *along with importance weights* $\{p_i\}_{i=1}^{n_L+K}$ *and evidence estimate* $\mathcal{Z}$.

---

associated with the likelihood: it describes the prior mass lying below the likelihood value $\psi$. The extreme values of the integrand, $\psi(X{=}0)$ and $\psi(X{=}1)$, therefore correspond to the minimum and maximum likelihood values found under the support of the prior $\pi(\theta)$.

Sampling proceeds in accordance with Algorithm 1. It begins with a random selection of a set of live points drawn from the prior. The number of live points, $n_L$, dictates the resolution at which the likelihood surface will be explored. At each iteration, a new sample is drawn from the prior and replaces the live point with the lowest likelihood, under the strict conditions that it possesses a higher likelihood. By following this iterative procedure the set of live points gradually converge upon regions of higher likelihood. The prior mass enclosed by these points shrinks exponentially. By the $jth$ iteration, the expectation of $X$ has reduced to $\langle X_j \rangle = [n_L/(n_L + 1)]^j X_0 \approx e^{-j/N} X_0$. Higher values of $n_L$ therefore come at the cost of a slower convergence rate. While the sequence of samples provides an estimate of $\mathcal{Z}$, an invaluable quantity in the context of Bayesian model selection, they also represent importance weighted samples of the posterior.

If we wished to perform inference in a low-dimensional setting, then a uniform sampling strategy would suffice. However since we wish to explore higher dimensions, we employ the PolyChord algorithm [Handley et al., 2015, Handley et al., 2015], which performs slice sampling [Neal, 2003]

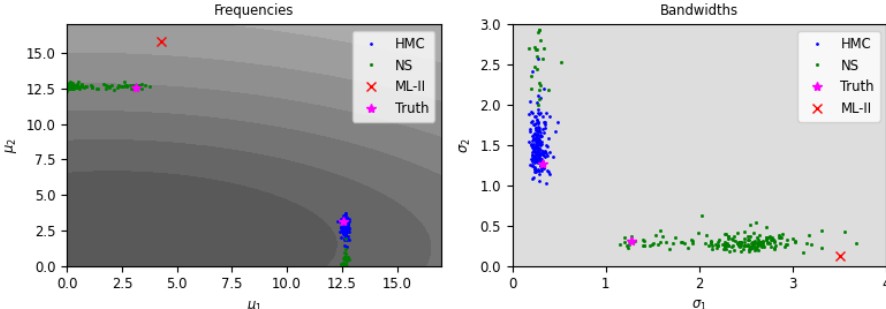

Figure 2: A comparison of methods for estimating the mean (left hand panel) and bandwidth (right hand panel) of two spectral components. In each case the ground truth is indicated by the magenta star, and the grey shading indicates the prior. We note that the nested sampling algorithm successfully samples from both modes, whereas HMC and ML-II can only identify a single mode.

at each iteration. Unless otherwise stated, we use 100 live points, which are bounded in a set of ellipsoids [Feroz et al., 2009]. These bounding surfaces allow the macroscopic structure of the likelihood contours to be traced, enabling a much more efficient sampling process. This approach has proven particularly adept at navigating multi-modal likelihood surfaces [Allison and Dunkley, 2014]. These attractive properties have motivated numerous scientific applications, including the detection of gravitational waves [Veitch et al., 2015], the categorisation of cosmic rays [Cholis et al., 2015], and the imaging of a supermassive black hole [Akiyama et al., 2019].

We implemented[4] the above algorithm via a combination of the DYNESTY [Speagle, 2020] and GPflow [De G. Matthews et al., 2017] software packages. For the former, we make use the 'rslice' sampling option provided by DYNESTY, along with the default number of five slices, and adopt the default criterion for convergence. This is defined as the point at which the estimated posterior mass contained within the set of live points falls below 1% of the total, specifically $\log(\hat{Z} + \Delta\hat{Z}) - \log(\hat{Z}) < 0.01$, where $\Delta\hat{Z} \simeq \psi_{max}X_i$.

## 4   Hyperpriors for the Spectral Mixture Kernel

In this section we outline a physically motivated set of priors for the hyperparameters which govern the spectral mixture kernel. As defined in (4), the three fundamental parameters of each spectral component are the mean frequency $\mu$, bandwidth $\sigma$, and weight $w$. For most of these hyperparameters, we find it is sufficient to impose a weakly informative prior of the form $\{\sigma, w, \sigma_n^2\} \sim \text{LogNormal}(0, 2)$ where $\sigma_n^2$ is the data noise variance. However the behaviour of the spectral component's characteristic frequency $\mu$ merits closer attention.

Two properties of the data strongly influence our perspective on which frequencies we can expect to observe: the fundamental frequency and the highest observable frequency. The fundamental frequency $\nu_F$ is the lowest frequency observable within the data, and is given by the inverse of the interval spanned by the observed $x$ locations. Meanwhile the maximum frequency $\nu_N$ represents the highest observable frequency. For gridded data, this is naturally determined by the Nyquist frequency, which is half the sampling frequency.

It is crucial to bear in mind that the spectral density we wish to model is that of the underlying process, and not the spectral density of the data. These two quantities are often very different, due to the limited scope of the observations. For example, the change in stock prices over the period of a single day cannot exhibit frequencies above $10^{-6}$Hz. Yet the process will have received contributions from long term fluctuations, such as those due to macroeconomic factors, whose periods can span many years. If we make no assumption regarding the relationship between the process we wish to model and the finite range over which it is observed, then a priori, some as-yet undiscovered frequency within the process ought to be considered equally likely to lie above or below the fundamental frequency.

---

[4]Code available at `https://github.com/frgsimpson/nsampling`

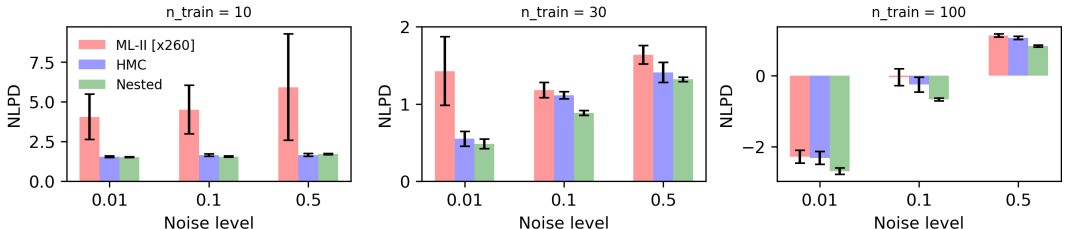

Figure 3: The impact of marginalising hyperparameters on the predictive performance (NLPD) of a spectral mixture kernel. The error bars at each noise level correspond to the mean NLPD and standard error of the mean across a diversity of latent functions. Nested sampling significantly outperforms ML-II in all nine experiments. The subplots contain the performance summaries under the different configurations. *Left:* Sparse distribution of training data ($n\_train = 10$). *Middle:* Moderate sized distribution of training data ($n\_train = 30$). *Right:* Dense distribution of training data ($n\_train = 100$) - all on a fixed domain [-1,+1].

Furthermore, given the large initial uncertainty in frequency $\mu$, it is appropriate to adopt a prior which spans many orders of magnitude.

Towards very low frequencies, $\mu \ll \nu_F$, a sinusoid contributes very little variance to the observations - an annual modulation makes a tiny contribution to a minute's worth of data. As we consider frequencies much lower than the fundamental frequency, it therefore becomes less likely that we will detect their contributions. We model this suppressed probability of observation with a broad Gaussian in $\log$ frequency for the regime $\mu < \nu_F$. Meanwhile, at frequencies above the Nyquist frequency, $\mu > \nu_F$, we encounter a degenerate behaviour: these sinusoids are indistinguishable from their counterparts at lower frequencies: they are said to be aliases of each other. As a result of this aliasing behaviour, the likelihood surface is littered with degenerate modes with identical likelihood values. From a computational perspective, it is advantageous to restrict our frequency parameter to a much narrower range than is permitted by our prior, while maintaining the same probability mass. As illustrated in the supplementary, mapping these higher frequencies down to their corresponding alias at $\mu < \nu_N$ yields a uniform prior on $\mu$.

$$\mu/\nu_F \sim \begin{cases} \text{Lognormal}(0,7), & \text{for } \mu < \nu_F \,, \\ \text{Uniform}(1, \nu_N/\nu_F), & \text{for } \nu_F < \mu < \nu_N \,. \end{cases} \tag{6}$$

## 5 Experimental Results

In this section we present results from a series of experiments on synthetic data in one and two dimensions, as well as real world time series data.

**Baselines.** For all of our experiments, the key benchmark we compare nested sampling to is the conventional ML-II method. This involves maximizing $\mathcal{L}(\theta) = \log p(\mathbf{y}|\theta)$ over the kernel hyperparameters, $\theta_\star = \arg\max_\theta \mathcal{L}(\theta)$. In addition to ML-II, it is also instructive to compare nested sampling against an MCMC method for marginalising the hyperparameters. Hamiltonian Monte Carlo [Duane et al., 1987] is a fundamental tool for inference in intractable Bayesian models. It relies upon gradient information to suppress random walk behaviour inherent to samplers like Metropolis-Hastings and its variants [for a detailed tutorial, see Neal et al., 2011]. In the experiments we use a self-tuning variant of HMC called the No-U-Turn Sampler [NUTS, Hoffman and Gelman, 2014] where the path length is adapted for every iteration. NUTS is frequently shown to work as well as a hand-tuned HMC; hence in this way we avoid the compute overhead in tuning for good values of the step-size ($\epsilon$) and path length ($L$). We use the version of NUTS available in the python package `pymc3`.

**Synthetic data.** As a first demonstration of how the different methods perform, we draw four samples from a two-component SM kernel with known weights, frequencies and bandwidths. For each latent function sample we construct noisy training data across three noise levels $[0.01, 0.1, 0.5]$ and three training sizes $[10, 30, 100]$, on a fixed input domain $[-1, 1]$. In this way we seek to quantify the quality of predictions and prediction uncertainty under weakly identified regimes characterised by very few data points in a fixed domain to strongly identified regimes characterized by a dense

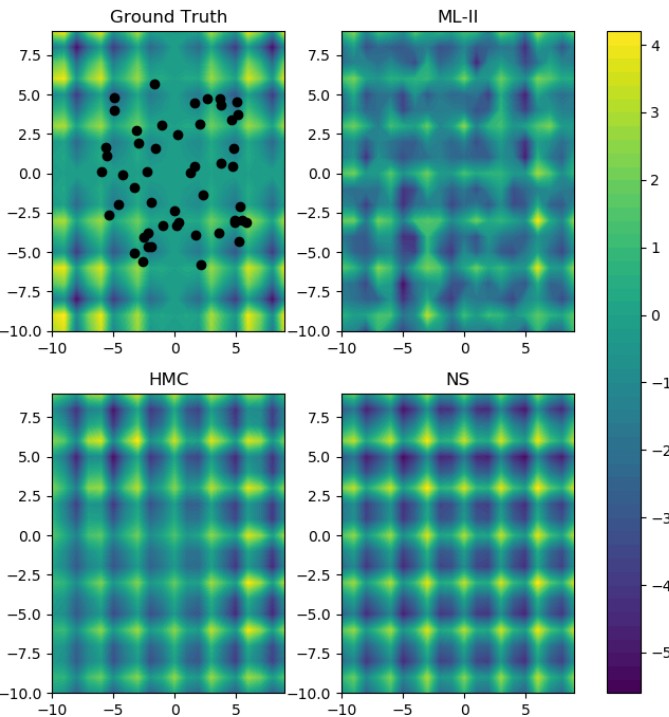

Figure 4: Learning a two-dimensional pattern with a 10 component spectral mixture kernel. *Top Left:* The ground truth, and the black dots denote the locations of the 50 training points. *Top Right:* Posterior predictive mean under ML-II. *Bottom Left:* Posterior predictive mean under HMC. *Bottom Right:* Posterior predictive mean under nested sampling.

distribution of data in a fixed domain. Further, the intrinsic noise level $\sigma_n^2$ of the data can also impact inference in weakly and strongly identified data regimes. In order to analyse the impact of $\sigma_n^2$ and training size we calculate the average performance across each of three different noise levels for each training set size.

We train under each of the three inference methods (ML-II, HMC, Nested) for each of the $9 \times 4$ data sets created and report prediction performance in terms of the average negative log predictive density (NLPD) in Figure 3. ML-II uses five random restarts with a initialisation protocol tied to the training data. Following protocols from Wilson and Adams [2013], the SM weights $(w_i)$ were initialised to the standard deviation of the targets $\mathbf{y}$ scaled by the number of components $(Q = 2)$. The SM bandwidths $(\sigma_i)$ were initialised to points randomly drawn from a truncated Gaussian $|\mathcal{N}(0, \max d(x, x')^2)|$ where $\max d(x, x')$ is the maximum distance between two training points and mean frequencies $(\mu_i)$ were drawn from $\mathrm{Unif}(0, \nu_N)$ to bound against degenerate frequencies. HMC used $\mathrm{LogNormal}(0, 2)$ priors for all the hyperparameters. The ML-II experiments used `gpytorch` while the HMC experiments used the NUTS sampler in `pymc3`.

The advantage of synethic data is we know the values we ought to recover. Figure 2 compares the three inference methods in their ability to recover the true hyperparameters for a synthetic dataset. This involved 100 datapoints, a noise amplitude of $\sigma_n = 0.1$ and a signal-to-noise ratio of $\approx 3.2$ on a fixed domain [-1,1]. The magenta star $\star$ denotes the true value and the red cross $\times$ denotes the ML-II estimate. The HMC and NS sampling schemes are both better at recovering the ground truth hyperparameters than the point estimates. Of particular significance, the nested sampling scheme is able to simultaneously sample from both modes inherent in the marginal likelihood.

In Figure 3 we look the impact on predictive performance, and notice that ML-II struggles with small training data sets, catastrophically underestimating the noise level (in the left hand panel the bars are rescaled for visibility). The two sampling methods, HMC and NS, perform comparably well when there are only 10 datapoints, but NS consistently outperforms HMC when there are 100 datapoints,

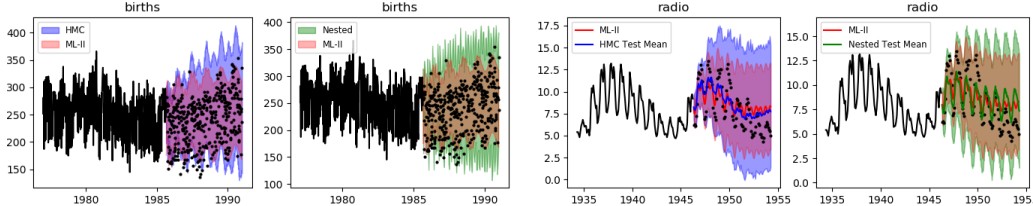

Figure 5: An illustration of the systematic underestimation of predictive uncertainty when adopting point estimates of kernel hyperparameters. Here we show $95\%$ test confidence intervals derived from the spectral mixture kernel. Black lines denote training data and the black dots denote test data. *Left:* 'BIRTHS' data set - the ML-II intervals capture $84\%$ of the test data where HMC and NS capture $93\%$ and $99\%$ of the test points respectively. *Right:* 'RADIO' data set (also showing test mean prediction) - the three methods show similar performance however, the average predictive density under test data is higher under the sampling methods.

across all noise levels. This may be due to its inherent ability to sample from multiple modes, as we saw in Figure 2.

**Two-dimensional Pattern Extrapolation.** To demonstrate how the inference methods adapt to higher dimensional problems, we revisit the challenge presented in Sun et al [26]. The two-dimensional ground truth function is given by $y = (\cos 2x_1 \times \cos 2x_2)\sqrt{|x_1 x_2|}$. This experiment marks a significant increase in the dimensionality of our parameters space, since in two dimensions each spectral component has five degrees of freedom. We train with 50 points chosen at random across $[-6, +6]$ in the $xy-$domain. The test points are defined on a $20 \times 20$ grid.

In Figure 4 we visualise the mean values of the posterior predictive distribution from three different inference methods: ML-II, HMC and nested sampling (NS). Visually, the reconstruction under the marginalised GP schemes (HMC / NS) appears to be superior to ML-II. Further, the $95\%$ confidence intervals (not visualised) differ markedly, as is evident from the NLPD values. These are given by 216, 2.56, and 2.62 for ML-II, HMC and NS respectively (*lower* is better). For reference we also trained the neural kernel network (Sun et al [26]), which achieved an NLPD of 3.8. Marginalised Gaussian processes comfortably outperform competing methods. ML-II was trained with Adam (learning rate=0.05) with 10 restarts and 2,000 iterations.

**Time series benchmarks.** To test of our inference methods on a selection of real world data, we evaluate their predictive performance against the thirteen 'industry' benchmark time series, as used by Lloyd et al. [2014][5]. The time series are of variable length, with up to $1,000$ data points in each.

[5]The raw data is available at
https://github.com/jamesrobertlloyd/gpss-research/tree/master/data/tsdlr-renamed

Table 1: NLPD values for various GP methods across a range of time series tasks.

| KERNEL | RBF | RBF | RBF | SPECTRAL | SPECTRAL | SPECTRAL | NKN |
|---|---|---|---|---|---|---|---|
| INFERENCE | ML-II | HMC | NS | ML-II | HMC | NS | ML-II |
| AIRLINE | 13 | 6.37 | 11.1 | 7.25 | 5.83 | **5.22** | 5.6 |
| BIRTHS | 5.27 | 5.26 | 5.26 | 5.17 | 5.23 | **4.96** | 5.42 |
| CALL CENTRE | 9.21 | 8.44 | 8.67 | 11 | 7.32 | **7.29** | 7.76 |
| GAS PRODUCTION | 14.9 | 12.39 | 18.8 | 15.2 | 12.4 | **11** | 12.4 |
| INTERNET | **11.2** | 11.35 | 11.4 | 11.3 | 11.4 | 13.2 | 12.6 |
| MAUNA | 2.53 | 2.64 | 2.53 | **1.5** | 3.32 | 1.8 | 3.4 |
| RADIO | 2.47 | 2.45 | 2.46 | 2.19 | 2.12 | **2.07** | 4.12 |
| SOLAR | 1.69 | 1.03 | 1.74 | 1.4 | 0.82 | **0.58** | 2.38 |
| SULPHURIC | 5.17 | 5.15 | 5.16 | 5.13 | 5.15 | **5.11** | 6.32 |
| TEMPERATURE | 2.79 | 2.81 | 2.8 | 2.8 | **2.49** | 2.52 | 4.2 |
| UNEMPLOYMENT | 11.9 | 11.03 | 11.3 | 12.8 | 11.2 | 10.6 | **8.5** |
| WAGES | 57.1 | 30.20 | 33.4 | 159 | 8.71 | 6.59 | **4.28** |
| WHEAT | 9.8 | 8.61 | 9.73 | 8.47 | 6.44 | 6.58 | **6.24** |
| MEAN | 11.31 | 8.29 | 10.2 | 18.7 | 6.34 | **5.96** | 6.40 |
|  |  |  |  | ±1.2 | ±0.03 | ±0.03 |  |

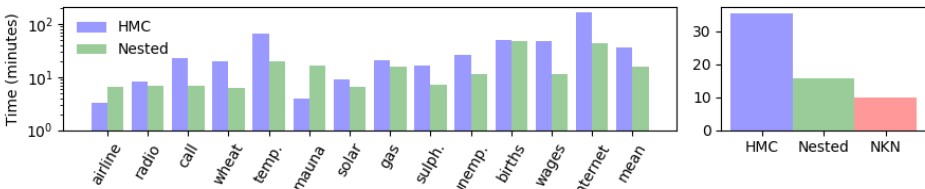

Figure 6: *Left:* Training time for each of the time series benchmarks. *Right:* Average training time across all data sets for the three inference methods we consider. Perhaps surprisingly, nested sampling is not substantially slower than NKN which is trained using ML-II.

All models are trained on normalised data, but the quoted test performance is evaluated in the original space. The output is scaled to zero mean and unit variance, while the input space is scaled to span the unit interval, thereby ensuring $\nu_F = 1$. Given the complex nature of the task, we utilise more spectral components than for the synthetic case. Our fiducial kernel has seven components ($Q = 7$), yielding a 22-dimensional hyperparameter space to explore. For reference, we also include results from the Neural Kernel Network [Sun et al., 2018] where a flexible kernel is learnt incrementally through a weighted composition of primitive base kernels. This is trained with the Adam optimiser for 100,000 iterations and a learning rate of $10^{-3}$.

In Table 1 we report the negative log predictive density (NLPD). The evaluation was conducted with a 60/40 train/test split. Each set of experiments was repeated three times with a different random seed (and used to generate the quoted uncertainties. For ML-II, each run included an initialisation protocol involving evaluating the marginal likelihood at one thousand points in the hyperparameter space, choosing the initial setting to the be the point with the highest marginal likelihood value.

We find that the spectral mixture kernel exhibits significant performance gains when using a sampling-based inference method compared to the conventional ML-II approach. The high average NLPD score for the 'WAGES' data-set underscores the lack of robustness associated with the ML-II approximation. We note that excluding this outlying score does not change the ranking of the methods. NS outperforms HMC on 11 of the 13 tasks, and also carries an advantage of faster evaluation times, and it provides an estimate of the model evidence (which could potentially be used to advise how many spectral components are necessary). While NS excels with the spectral mixture kernel, it cannot offer similar gains with the simpler RBF model. We again interpret this as being due to its ability to traverse the multi-modal likelihood surface better than other methods. Examples of the posterior distributions are illustrated in Figure 5, while more detailed results for various configurations of the nested sampler can be found in the Appendix.

While Nested Sampling is a lesser known inference technique, it is one which has gained a dramatic increase in usage within the physics community over the past decade. We note that much the same could have been said for HMC in the early 1990s. The key advantage of NS is that the location of the next sample is not governed by the single previous point (as in MCMC), but by whole clouds of 'live points'. Hence it readily samples across multiple modes in the marginal likelihood surface, as can be seen in Figure 2. The hyperparameter samples obtained from NS induce a greater diversity of functions than those from HMC. This also sheds light on the question about the uncertainty intervals, NS samples give wider intervals in prediction tasks as their samples are more diverse, and these more rigorous uncertainty estimates led to improved test performance.

**Computation.** Figure 6 depicts the wall clock time required for training during our time-series experiments, all of which utilised a single Nvidia GTX1070 GPU. For HMC this corresponds to a single chain with 500 warm-up iterations and 500 samples. Across these benchmarks, HMC was found to be slower by approximately a factor of two compared to nested sampling. While sampling hyperparameters will add some computational overhead, there are many means by which the marginal likelihood could be computed more rapidly. This includes inducing point methods (sparse GPs) and structure exploiting Kronecker methods when the data lie on a grid. Overall, the nested sampling approach was not found to be significantly slower than sophisticated kernel learning schemes like the NKN kernel learning method with ML-II inference.

**Consequences.** What are the potential societal impacts? By developing inference methods with a more reliable quantification of uncertainty, this may lead to safer decision making. On the other hand, greater computational overheads can be accompanied with a greater carbon footprint.

## 6    Discussion

We present a first application of nested sampling to Gaussian processes, previously only performed in the context of astronomy. It offers a promising alternative to established GP inference methods, given its inherent capability of exploring multimodal surfaces, especially since modern incarnations such as PolyChord [Handley et al., 2015] enable higher dimensional spaces ($D \lesssim 50$) to be explored.

The spectral mixture kernel possesses a challenging likelihood surface, which made it an excellent target to showcase the versatility of nested sampling. We have performed the first demonstration of marginalised spectral mixture kernels, finding significant performance gains are readily available when using either HMC or NS, relative to the conventional approach of using point estimates for the hyperparameters. The benefits of performing full GP marginalisation are three-fold. Firstly, it defends against overfitting of hyperparameters, which may occur when their uncertainty is substantial. Secondly, there is a lower susceptibility to getting trapped in sub-optimal local minima. Thirdly, they provide more robust prediction intervals by accounting for hyperparameter uncertainty. These three advantages are most pronounced when many hyperparameters are present, or the data is noisy.

While a pathological marginal likelihood geometry can pose problems for both gradient based optimisation and sampling; sampling schemes are able to quantify them better if the practical difficulties of deploying them are overcome. The nested sampling scheme, which does not require gradient information, provides an opportunity to sample the multimodal posterior at a fraction of the cost of running HMC. Further, it is crucial to ask if we are compromising the full expressivity of kernels by resorting to point estimates in the learning exercise? A marginalised GP with a standard SM kernel was found to improve considerably upon its conventional counterpart. We also note that the methods discussed here could be generalised to operate in a sparse variational GP framework when dealing with larger datasets. Future work will focus in this direction.

### Acknowledgments and Disclosure of Funding

The authors would like to thank João Faria, Joshua Albert, Elvijs Sarkans, and the anonymous reviewers for helpful comments. VL acknowledges funding from the Alan Turing Institute and Qualcomm Innovation Fellowship (Europe).

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
