# Supplementary Material for Marginalised Gaussian Processes with Nested Sampling

## 1 Experimental Results

This section presents a more detailed description and results from our three sets of experiments: ground truth recovery, synthetic data, and realistic time series.

**Ground Truth Recovery** In this subsection we summarise the performance of ML-II, HMC and Nested sampling inference in recovering the true setting of the hyperparameters under two different noise settings, for 100 training data on a fixed domain [-1,1].

The top row of panels in Fig.1 indicate a low noise setting $\sigma_n = 0.01$ and the bottom row indicates a higher noise setting of $\sigma_n = 0.1$. The magenta star $\star$ denotes the true value and the red cross $\times$ denotes the ML-II estimate. The sampling schemes HMC and Nested are both better at recovering the target than the point estimate. While ML-II estimates the noise-level to a high precision when the noise is low (top-row, $\sigma_n = 0.01$), it does not fare so well when noise level is raised by an order of magnitude (bottom row, $\sigma_n = 0.1$). In this case, the estimate of the intrinsic noise level is off by several orders of magnitude. The sampling schemes prove to be far more robust in recovering the frequencies, bandwidths and noise level, especially when operating in the low signal-to-noise regime.

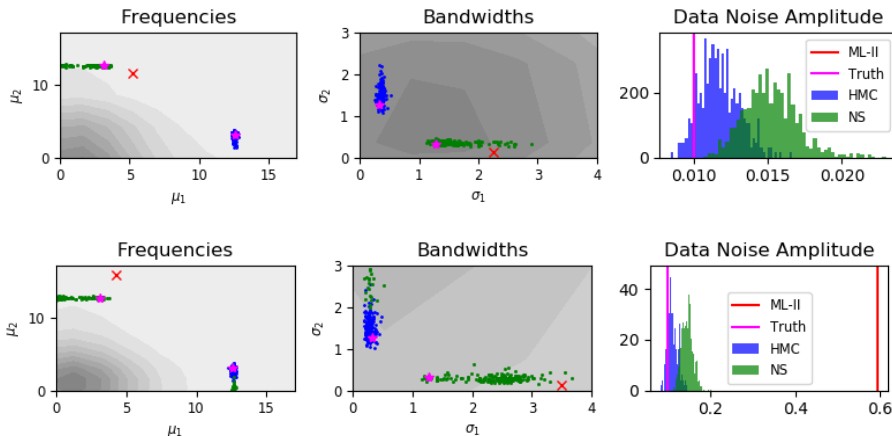

Figure 1: A comparison of hyperparameter estimation, where the ground truth is indicated by the magenta star. The grey shading indicates the prior. *Left:* Recovering the mean frequency parameters of the two spectral components. *Middle:* Recovering the two bandwidth parameters of the two spectral components. *Right:* Recovering the data noise level ($\sigma_n$). The true hyperparameters are $[\mu_1, \mu_2] = [3.14, 12.56]$ and $[\sigma_1, \sigma_2] = [1.27, 0.32]$. For frequencies and bandwidths we note the symmetry i.e. the estimates can converge on $[\mu_1, \mu_2]$ or $[\mu_2, \mu_1]$, and that the nested sampling algorithm successfully identifies both.

**Synthetic Data**    Fig.2 shows the test mean squared error across the three inference schemes. The sampling schemes largely dominate the ML-II method when the hyperparameters are well identified ($n\_train = 100$). The data generating configurations are the same as the ones described in the main paper.

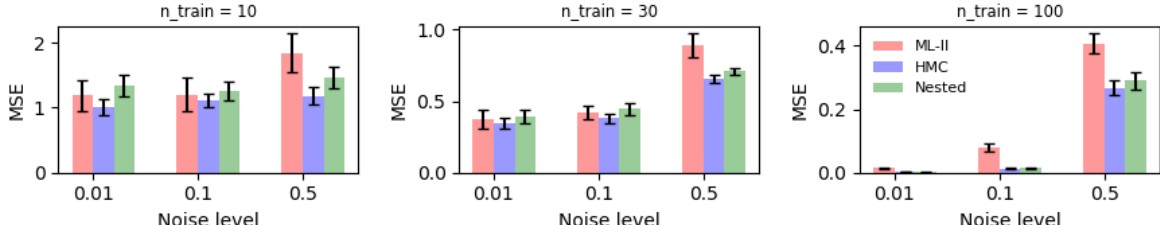

Figure 2: Mean-squared error for synthetic data sets under different noise levels and training set sizes.

**Time Series**    Here we present further results from the benchmark time series experiments. Instead of making full use of the data, we consider only the first 100 points as training data, followed by testing with the subsequent 30 points. As with the results from the full training set, significant performance gains are found when marginalising the hyperparameters of the spectral mixture kernels. However in this case, the nested sampling algorithm doesn't offer a performance advantage over HMC. We speculate this may be due to the simpler likelihood surface associated with the smaller set of training data. Fewer modes in the surface would facilitate exploration via HMC.

## 2   Spectral Priors

Figure 3 shows the parameter space for the frequency and bandwidth of a single spectral component. The likelihood surfaces adjacent to any of the dashed lines are mirror images of each other. It is therefore preferable to avoid exploring multiple copies of these regions when performing nested sampling, as it will attempt to locate the duplicate modes, dispersing the live points.

To give a clearer picture of how the parameters of the spectral mixture kernel are inferred via nested sampling, we take as a example the radio experiment. The posterior distribution in this case can

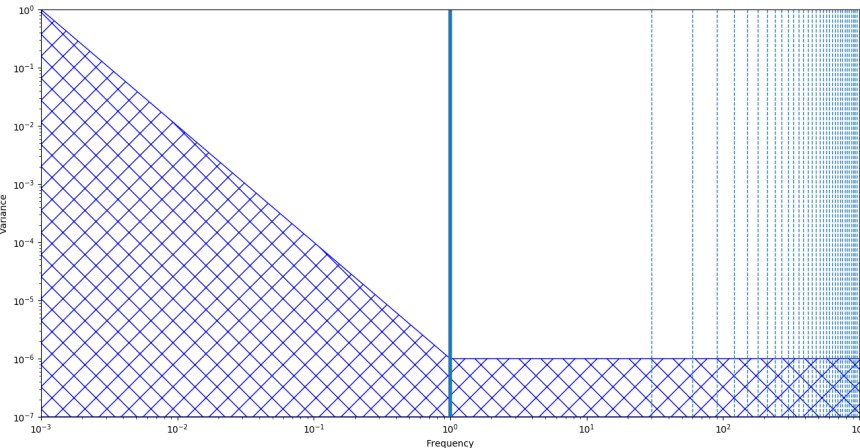

Figure 3: Schematic of the observability of a spectral component as a function of frequency and variance. The fundamental frequency is denoted by the solid vertical line, while dashed vertical lines indicate multiples of the Nyquist frequency. The hatched region denotes the regime where the variance is deemed too low to be observed.

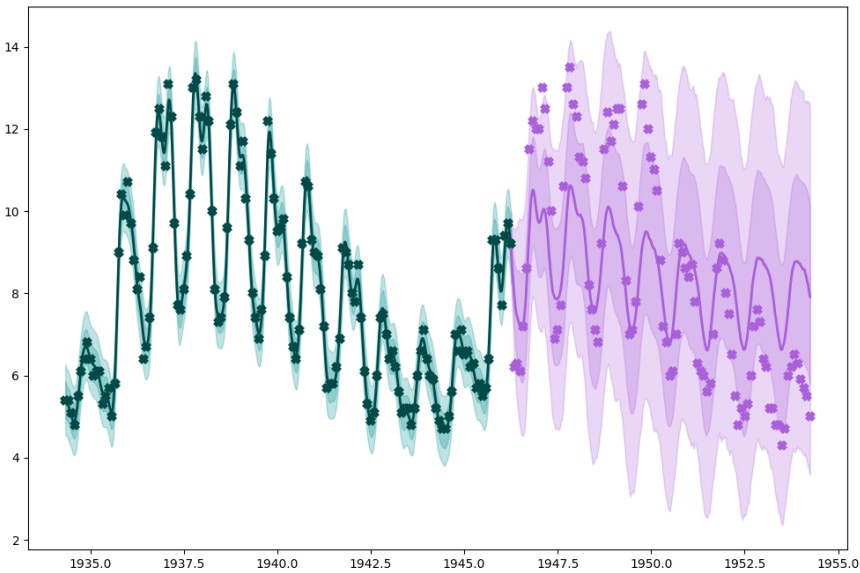

Figure 4: 68 and 95 per cent confidence intervals for the radio dataset, derived from nested sampling of a seven-component spectral mixture kernel.

be seen in Figure 4. The corresponding joint posterior distribution of the 22 hyperparameters are displayed in Fig.5.

We make use of the dynesty package in order to perform nested sampling. The full set of configuration parameters used can be found in Table 1.

Table 1: A summary of the configuration settings used to perform nested sampling. Most of these adhere to the default set-up in dynesty, with the most significant changes being in the sampling method, and a reduction in the number of live points.

|             | Fiducial | Default |
|-------------|----------|---------|
| method      | rslice   | auto    |
| live points | 100      | 500     |
| Bound       | multi    | multi   |
| slices      | 5        | 5       |
| dlogz       | 0.01     | 0.01    |
| max iter    | None     | None    |
| max call    | None     | None    |
| min eff     | 3        | 10      |
| vol dec     | 0.5      | 0.5     |

**Time-Series**    The tables below summarize the posterior samples and sampler statistics based on the trace containing joint samples from the HMC run. The columns hpd_2.5 and hpd_97.5 calculate the highest posterior density interval based on marginal posteriors. $\text{n\_eff} = \dfrac{MN}{1 + 2\sum_{t=1}^{T}\hat{\rho}_t}$ computes effective sample size where $M$ is the number of chains, $N$ is the number of samples in each chain and $\rho_t$ denotes auto-correlation at lag $t$. For the results reported below $N = 500$ and $M = 1$. Each chain was run with 500 warm-up iterations for the sampler to adapt to an optimal step-size. The 22 hyperparameters sampled are the mean frequencies, bandwidths and weights $\{\text{mu[i], bw[i], w[i]}\}_{i=0}^{6}$

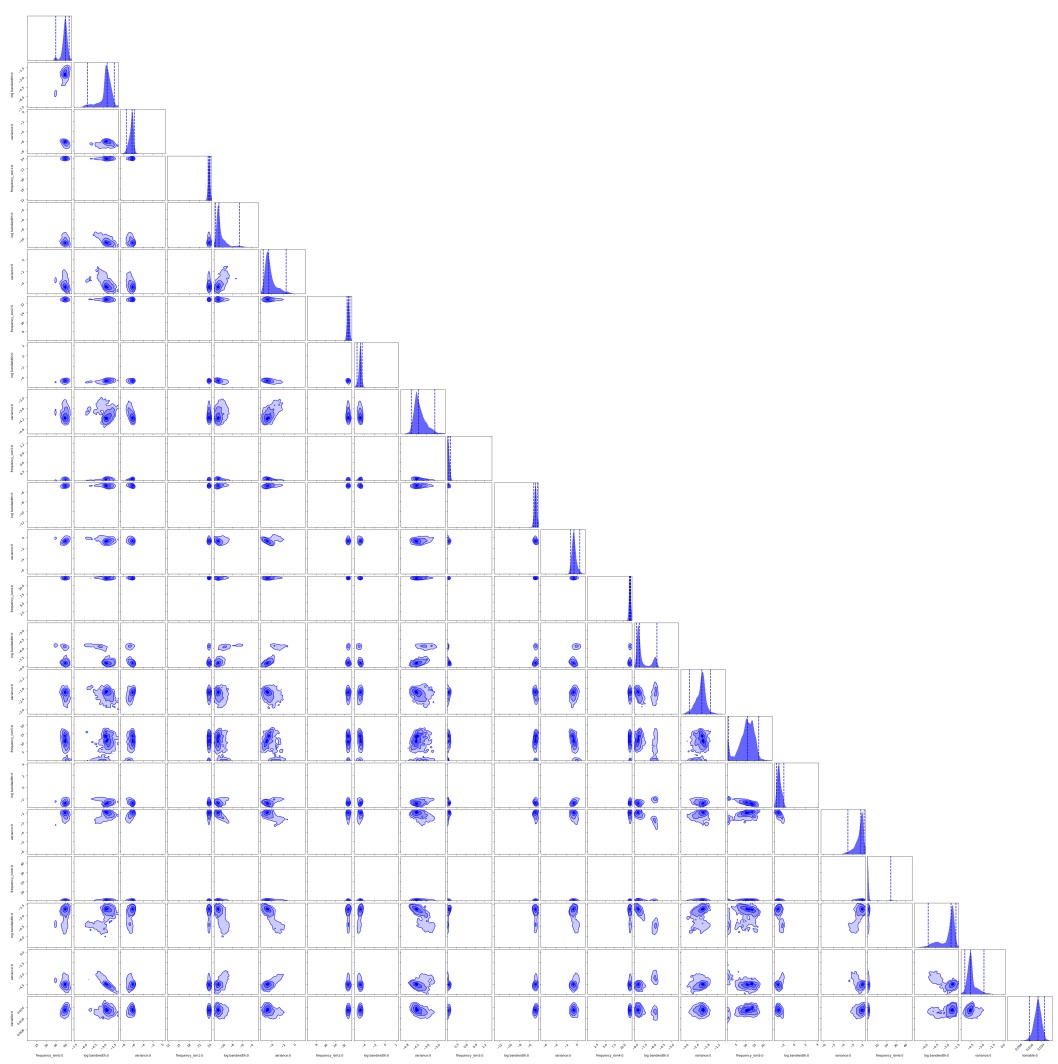

Figure 5: Joint posterior distributions for the 22 hyperparameters associated with Figure 4.

for a 7 component spectral mixture kernel. The hyperparameter $n$ denotes the noise standard deviation $\sigma_n$.

| | Hyperparameter | mean | sd | hpd_2.5 | hpd_97.5 | mc_error | n_eff |
|---|---|---|---|---|---|---|---|
| | mu[0] | 5.441 | 12.788 | 0.003 | 27.993 | 4.533 | 10.0 |
| | mu[1] | 0.375 | 0.554 | 0.002 | 1.155 | 0.05 | 23.0 |
| | mu[2] | 14.32 | 0.178 | 14.036 | 14.683 | 0.046 | 27.0 |
| | mu[3] | 2.749 | 7.062 | 0.033 | 25.552 | 2.913 | 12.0 |
| | mu[4] | 6.939 | 0.151 | 6.701 | 7.211 | 0.033 | 22.0 |
| | mu[5] | 0.596 | 1.298 | 0.02 | 1.797 | 0.137 | 42.0 |
| | mu[6] | 1.252 | 6.518 | 0.003 | 2.238 | 0.826 | 51.0 |
| | bw[0] | 0.69 | 5.153 | 0.002 | 1.213 | 0.256 | 58.0 |
| | bw[1] | 1.309 | 9.737 | 0.003 | 1.269 | 0.976 | 239.0 |
| | bw[2] | 0.125 | 0.111 | 0.003 | 0.351 | 0.014 | 15.0 |
| **Airline** | bw[3] | 1.236 | 3.694 | 0.014 | 3.213 | 0.314 | 8.0 |
| | bw[4] | 0.177 | 0.092 | 0.027 | 0.309 | 0.021 | 22.0 |
| | bw[5] | 0.459 | 0.685 | 0.004 | 1.313 | 0.072 | 42.0 |
| | bw[6] | 6.0 | 25.438 | 0.003 | 29.574 | 5.349 | 12.0 |
| | w[0] | 0.829 | 2.259 | 0.001 | 3.031 | 0.436 | 3.0 |
| | w[1] | 1.815 | 6.441 | 0.009 | 5.089 | 0.352 | 38.0 |
| | w[2] | 0.202 | 0.4 | 0.009 | 0.575 | 0.041 | 20.0 |
| | w[3] | 1.084 | 3.843 | 0.002 | 4.105 | 0.366 | 9.0 |
| | w[4] | 1.223 | 7.289 | 0.048 | 2.011 | 0.627 | 245.0 |
| | w[5] | 3.423 | 6.384 | 0.008 | 17.871 | 1.976 | 15.0 |
| | w[6] | 1.929 | 3.752 | 0.002 | 9.162 | 0.993 | 5.0 |
| | n | 0.155 | 0.035 | 0.088 | 0.211 | 0.008 | 27.0 |

**Solar**

| Hyperparameter | mean | sd | hpd_2.5 | hpd_97.5 | mc_error | n_eff |
|---|---|---|---|---|---|---|
| mu[0] | 1.203 | 2.736 | 0.001 | 3.387 | 0.288 | 142.0 |
| mu[1] | 4.683 | 8.516 | 0.003 | 22.88 | 2.765 | 11.0 |
| mu[2] | 2.105 | 4.909 | 0.006 | 7.927 | 0.555 | 135.0 |
| mu[3] | 2.275 | 5.065 | 0.002 | 8.911 | 1.148 | 33.0 |
| mu[4] | 1.746 | 3.948 | 0.001 | 4.9 | 0.738 | 125.0 |
| mu[5] | 1.812 | 4.58 | 0.002 | 4.037 | 0.695 | 127.0 |
| mu[6] | 23.475 | 1.726 | 20.725 | 27.323 | 0.28 | 61.0 |
| bw[0] | 1.897 | 6.4 | 0.005 | 3.111 | 0.649 | 96.0 |
| bw[1] | 3.812 | 8.317 | 0.001 | 25.13 | 1.349 | 26.0 |
| bw[2] | 7.012 | 19.894 | 0.007 | 29.453 | 5.351 | 11.0 |
| bw[3] | 9.663 | 14.208 | 0.002 | 32.005 | 9.512 | 3.0 |
| bw[4] | 6.376 | 12.311 | 0.005 | 29.365 | 4.035 | 16.0 |
| bw[5] | 1.265 | 2.48 | 0.001 | 3.064 | 0.324 | 120.0 |
| bw[6] | 4.002 | 3.254 | 0.768 | 9.662 | 0.689 | 25.0 |
| w[0] | 0.572 | 1.021 | 0.002 | 1.874 | 0.062 | 144.0 |
| w[1] | 0.67 | 1.459 | 0.001 | 2.286 | 0.148 | 111.0 |
| w[2] | 0.354 | 0.653 | 0.0 | 1.318 | 0.043 | 80.0 |
| w[3] | 0.467 | 0.911 | 0.004 | 1.877 | 0.138 | 83.0 |
| w[4] | 0.617 | 2.012 | 0.002 | 1.634 | 0.133 | 28.0 |
| w[5] | 0.738 | 2.445 | 0.0 | 2.079 | 0.112 | 197.0 |
| w[6] | 0.166 | 0.07 | 0.066 | 0.32 | 0.008 | 48.0 |
| n | 0.226 | 0.027 | 0.182 | 0.273 | 0.004 | 43.0 |

**Mauna**

| Hyperparameter | mean | sd | hpd_2.5 | hpd_97.5 | mc_error | n_eff |
|---|---|---|---|---|---|---|
| mu[0] | 0.323 | 0.638 | 0.003 | 0.819 | 0.07 | 270.0 |
| mu[1] | 0.259 | 0.271 | 0.005 | 0.675 | 0.018 | 254.0 |
| mu[2] | 0.264 | 0.232 | 0.002 | 0.718 | 0.011 | 298.0 |
| mu[3] | 0.247 | 0.263 | 0.0 | 0.685 | 0.012 | 429.0 |
| mu[4] | 0.257 | 0.255 | 0.003 | 0.722 | 0.011 | 440.0 |
| mu[5] | 0.269 | 0.272 | 0.0 | 0.765 | 0.013 | 440.0 |
| mu[6] | 0.245 | 0.212 | 0.003 | 0.639 | 0.009 | 363.0 |
| bw[0] | 0.185 | 0.218 | 0.004 | 0.534 | 0.01 | 528.0 |
| bw[1] | 0.196 | 0.282 | 0.0 | 0.473 | 0.016 | 360.0 |
| bw[2] | 0.172 | 0.166 | 0.002 | 0.478 | 0.007 | 376.0 |
| bw[3] | 0.183 | 0.232 | 0.001 | 0.522 | 0.01 | 531.0 |
| bw[4] | 0.18 | 0.197 | 0.001 | 0.535 | 0.009 | 309.0 |
| bw[5] | 0.171 | 0.15 | 0.002 | 0.452 | 0.007 | 467.0 |
| bw[6] | 0.179 | 0.175 | 0.002 | 0.477 | 0.008 | 420.0 |
| w[0] | 1.532 | 3.161 | 0.001 | 6.289 | 0.166 | 278.0 |
| w[1] | 1.269 | 2.35 | 0.0 | 5.433 | 0.109 | 346.0 |
| w[2] | 1.38 | 3.057 | 0.001 | 5.211 | 0.198 | 406.0 |
| w[3] | 2.197 | 8.752 | 0.001 | 7.209 | 0.41 | 470.0 |
| w[4] | 1.878 | 5.825 | 0.002 | 6.786 | 0.297 | 496.0 |
| w[5] | 1.568 | 4.4 | 0.002 | 5.362 | 0.21 | 478.0 |
| w[6] | 1.517 | 3.262 | 0.0 | 6.029 | 0.138 | 421.0 |
| n | 0.219 | 0.009 | 0.203 | 0.237 | 0.0 | 463.0 |

**Wheat**

| Hyperparameter | mean | sd | hpd_2.5 | hpd_97.5 | mc_error | n_eff |
|---|---|---|---|---|---|---|
| mu[0] | 4.383 | 9.584 | 0.002 | 23.762 | 1.981 | 35.0 |
| mu[1] | 3.528 | 5.881 | 0.002 | 14.145 | 1.43 | 26.0 |
| mu[2] | 4.223 | 6.242 | 0.002 | 16.112 | 1.321 | 48.0 |
| mu[3] | 2.584 | 5.283 | 0.001 | 14.086 | 1.306 | 83.0 |
| mu[4] | 2.262 | 4.318 | 0.002 | 13.331 | 1.41 | 29.0 |
| mu[5] | 1.556 | 3.831 | 0.002 | 5.861 | 0.346 | 188.0 |
| mu[6] | 2.844 | 4.772 | 0.006 | 13.514 | 2.003 | 14.0 |
| bw[0] | 13.452 | 17.514 | 0.002 | 42.654 | 10.143 | 3.0 |
| bw[1] | 6.74 | 12.879 | 0.006 | 34.173 | 3.169 | 38.0 |
| bw[2] | 8.622 | 42.731 | 0.003 | 33.009 | 3.253 | 56.0 |
| bw[3] | 6.486 | 13.446 | 0.004 | 34.901 | 2.667 | 33.0 |
| bw[4] | 1.689 | 5.909 | 0.002 | 3.659 | 0.573 | 119.0 |
| bw[5] | 12.765 | 17.856 | 0.001 | 40.11 | 11.495 | 3.0 |
| bw[6] | 2.688 | 7.035 | 0.004 | 13.883 | 0.804 | 110.0 |
| w[0] | 0.444 | 1.161 | 0.001 | 1.848 | 0.139 | 67.0 |
| w[1] | 0.657 | 2.901 | 0.001 | 2.495 | 0.133 | 135.0 |
| w[2] | 0.625 | 1.675 | 0.002 | 2.855 | 0.1 | 179.0 |
| w[3] | 0.397 | 0.868 | 0.0 | 1.3 | 0.111 | 61.0 |
| w[4] | 0.771 | 2.275 | 0.004 | 2.92 | 0.126 | 236.0 |
| w[5] | 0.708 | 2.032 | 0.002 | 2.381 | 0.155 | 107.0 |
| w[6] | 0.838 | 2.728 | 0.002 | 3.272 | 0.157 | 136.0 |
| n | 0.129 | 0.053 | 0.021 | 0.199 | 0.006 | 92.0 |

**Temperature**

| Hyperparameter | mean | sd | hpd_2.5 | hpd_97.5 | mc_error | n_eff |
|---|---|---|---|---|---|---|
| mu[0] | 3.109 | 6.892 | 0.005 | 11.861 | 0.989 | 5.0 |
| mu[1] | 2.38 | 4.257 | 0.004 | 11.02 | 0.576 | 43.0 |
| mu[2] | 5.926 | 0.072 | 5.772 | 6.045 | 0.007 | 89.0 |
| mu[3] | 4.169 | 13.09 | 0.007 | 11.535 | 0.759 | 43.0 |
| mu[4] | 16.579 | 24.229 | 0.049 | 65.489 | 16.668 | 4.0 |
| mu[5] | 5.897 | 9.641 | 0.036 | 11.901 | 2.623 | 4.0 |
| mu[6] | 4.71 | 10.925 | 0.005 | 11.537 | 2.075 | 17.0 |
| bw[0] | 14.533 | 35.555 | 0.014 | 91.228 | 14.155 | 4.0 |
| bw[1] | 8.093 | 19.723 | 0.002 | 45.652 | 2.39 | 29.0 |
| bw[2] | 0.056 | 0.064 | 0.002 | 0.173 | 0.029 | 3.0 |
| bw[3] | 374.1 | 56.544 | 265.853 | 439.078 | 30.709 | 4.0 |
| bw[4] | 28.195 | 101.687 | 0.035 | 92.316 | 10.997 | 3.0 |
| bw[5] | 7.815 | 25.062 | 0.008 | 43.333 | 3.286 | 4.0 |
| bw[6] | 5.504 | 18.456 | 0.003 | 18.906 | 1.731 | 29.0 |
| w[0] | 0.097 | 0.279 | 0.001 | 0.224 | 0.022 | 29.0 |
| w[1] | 0.168 | 0.58 | 0.002 | 0.42 | 0.039 | 43.0 |
| w[2] | 1.026 | 1.325 | 0.146 | 2.456 | 0.158 | 51.0 |
| w[3] | 0.338 | 0.037 | 0.258 | 0.404 | 0.004 | 93.0 |
| w[4] | 0.11 | 0.234 | 0.002 | 0.342 | 0.016 | 74.0 |
| w[5] | 0.16 | 0.462 | 0.001 | 0.533 | 0.036 | 30.0 |
| w[6] | 0.234 | 0.618 | 0.001 | 1.037 | 0.075 | 51.0 |
| n | 0.128 | 0.116 | 0.003 | 0.325 | 0.069 | 9.0 |

**Internet**

| Hyperparameter | mean | sd | hpd_2.5 | hpd_97.5 | mc_error | n_eff |
|---|---|---|---|---|---|---|
| mu[0] | 41.251 | 0.164 | 40.951 | 41.573 | 0.007 | 499.0 |
| mu[1] | 1.77 | 2.182 | 0.004 | 5.986 | 0.473 | 34.0 |
| mu[2] | 171.99 | 10.505 | 157.377 | 188.833 | 9.412 | 1.0 |
| mu[3] | 6.61 | 11.249 | 0.003 | 35.521 | 3.684 | 27.0 |
| mu[4] | 1.641 | 3.151 | 0.002 | 6.501 | 0.344 | 71.0 |
| mu[5] | 82.101 | 0.797 | 79.981 | 83.175 | 0.311 | 13.0 |
| mu[6] | 46.981 | 0.219 | 46.621 | 47.455 | 0.014 | 364.0 |
| bw[0] | 0.316 | 0.102 | 0.161 | 0.52 | 0.006 | 292.0 |
| bw[1] | 2.425 | 2.307 | 0.008 | 6.059 | 0.381 | 38.0 |
| bw[2] | 31.925 | 9.213 | 17.522 | 48.392 | 7.874 | 2.0 |
| bw[3] | 36.062 | 10.448 | 12.062 | 48.814 | 9.154 | 1.0 |
| bw[4] | 2.481 | 4.185 | 0.002 | 6.395 | 1.324 | 9.0 |
| bw[5] | 1.794 | 1.53 | 0.096 | 4.191 | 1.465 | 1.0 |
| bw[6] | 0.215 | 0.413 | 0.005 | 0.478 | 0.04 | 182.0 |
| w[0] | 0.573 | 0.708 | 0.052 | 1.85 | 0.065 | 257.0 |
| w[1] | 0.685 | 1.551 | 0.023 | 1.712 | 0.112 | 80.0 |
| w[2] | 0.009 | 0.002 | 0.006 | 0.012 | 0.001 | 2.0 |
| w[3] | 0.116 | 0.027 | 0.063 | 0.165 | 0.004 | 44.0 |
| w[4] | 0.752 | 1.515 | 0.015 | 2.814 | 0.096 | 75.0 |
| w[5] | 0.098 | 0.121 | 0.016 | 0.26 | 0.034 | 125.0 |
| w[6] | 0.198 | 0.408 | 0.007 | 0.529 | 0.025 | 359.0 |
| n | 0.051 | 0.004 | 0.045 | 0.059 | 0.001 | 24.0 |

**Call centre**

| Hyperparameter | mean | sd | hpd_2.5 | hpd_97.5 | mc_error | n_eff |
|---|---|---|---|---|---|---|
| mu[0] | 5.997 | 7.882 | 0.005 | 17.992 | 3.029 | 11.0 |
| mu[1] | 2.364 | 3.009 | 0.004 | 9.135 | 1.19 | 5.0 |
| mu[2] | 4.539 | 4.103 | 0.007 | 9.503 | 1.899 | 11.0 |
| mu[3] | 1.236 | 2.341 | 0.003 | 3.874 | 0.285 | 64.0 |
| mu[4] | 21.956 | 10.027 | 0.14 | 27.02 | 5.682 | 7.0 |
| mu[5] | 1.139 | 1.933 | 0.001 | 3.801 | 0.403 | 34.0 |
| mu[6] | 0.707 | 1.614 | 0.005 | 2.178 | 0.168 | 33.0 |
| bw[0] | 2.164 | 5.396 | 0.002 | 9.925 | 0.537 | 22.0 |
| bw[1] | 1.591 | 2.024 | 0.001 | 6.096 | 0.801 | 18.0 |
| bw[2] | 3.397 | 6.896 | 0.006 | 19.817 | 3.386 | 11.0 |
| bw[3] | 4.497 | 8.251 | 0.002 | 20.325 | 1.67 | 22.0 |
| bw[4] | 1.351 | 5.237 | 0.001 | 3.323 | 1.416 | 4.0 |
| bw[5] | 0.934 | 1.577 | 0.002 | 3.587 | 0.319 | 22.0 |
| bw[6] | 4.642 | 10.421 | 0.003 | 28.471 | 2.406 | 30.0 |
| w[0] | 0.6 | 4.572 | 0.001 | 2.309 | 0.226 | 19.0 |
| w[1] | 0.565 | 1.936 | 0.001 | 1.916 | 0.257 | 25.0 |
| w[2] | 0.368 | 0.958 | 0.003 | 1.844 | 0.103 | 21.0 |
| w[3] | 0.949 | 3.11 | 0.004 | 4.606 | 0.315 | 22.0 |
| w[4] | 0.188 | 0.828 | 0.003 | 0.435 | 0.075 | 19.0 |
| w[5] | 0.817 | 1.771 | 0.001 | 3.043 | 0.159 | 21.0 |
| w[6] | 1.592 | 5.663 | 0.004 | 5.312 | 0.558 | 31.0 |
| n | 0.104 | 0.037 | 0.028 | 0.155 | 0.007 | 32.0 |

**Radio**

| Hyperparameter | mean | sd | hpd_2.5 | hpd_97.5 | mc_error | n_eff |
|---|---|---|---|---|---|---|
| mu[0] | 2.133 | 4.166 | 0.004 | 9.105 | 0.505 | 132.0 |
| mu[1] | 35.677 | 0.329 | 35.074 | 36.331 | 0.014 | 544.0 |
| mu[2] | 1.206 | 2.126 | 0.004 | 2.306 | 0.189 | 259.0 |
| mu[3] | 2.261 | 4.327 | 0.006 | 10.189 | 0.635 | 61.0 |
| mu[4] | 24.16 | 0.936 | 22.76 | 26.284 | 0.099 | 119.0 |
| mu[5] | 11.856 | 0.322 | 11.308 | 12.461 | 0.025 | 242.0 |
| mu[6] | 4.822 | 14.166 | 0.003 | 23.511 | 2.629 | 28.0 |
| bw[0] | 3.814 | 8.143 | 0.001 | 22.917 | 1.724 | 42.0 |
| bw[1] | 0.312 | 0.356 | 0.001 | 0.935 | 0.024 | 225.0 |
| bw[2] | 12.055 | 201.131 | 0.004 | 23.783 | 9.925 | 25.0 |
| bw[3] | 7.647 | 11.432 | 0.004 | 31.073 | 5.255 | 7.0 |
| bw[4] | 1.571 | 1.582 | 0.002 | 3.603 | 0.29 | 46.0 |
| bw[5] | 0.644 | 0.211 | 0.303 | 1.099 | 0.014 | 309.0 |
| bw[6] | 7.423 | 15.978 | 0.005 | 32.507 | 3.135 | 15.0 |
| w[0] | 0.605 | 1.691 | 0.0 | 1.91 | 0.088 | 61.0 |
| w[1] | 0.068 | 0.219 | 0.003 | 0.191 | 0.013 | 264.0 |
| w[2] | 0.525 | 0.908 | 0.002 | 1.795 | 0.047 | 51.0 |
| w[3] | 0.513 | 1.382 | 0.002 | 1.744 | 0.11 | 44.0 |
| w[4] | 0.065 | 0.09 | 0.004 | 0.152 | 0.006 | 299.0 |
| w[5] | 0.418 | 0.367 | 0.096 | 0.944 | 0.024 | 320.0 |
| w[6] | 0.512 | 1.849 | 0.001 | 1.943 | 0.13 | 10.0 |
| n | 0.127 | 0.045 | 0.01 | 0.179 | 0.015 | 10.0 |

**Gas Production**

| Hyperparameter | mean | sd | hpd_2.5 | hpd_97.5 | mc_error | n_eff |
|---|---|---|---|---|---|---|
| mu[0] | 0.42 | 0.436 | 0.001 | 1.189 | 0.023 | 259.0 |
| mu[1] | 3.401 | 8.313 | 0.003 | 15.849 | 1.277 | 4.0 |
| mu[2] | 0.59 | 0.973 | 0.005 | 1.684 | 0.067 | 243.0 |
| mu[3] | 0.497 | 0.638 | 0.006 | 1.403 | 0.043 | 239.0 |
| mu[4] | 0.627 | 0.909 | 0.009 | 2.232 | 0.117 | 77.0 |
| mu[5] | 23.603 | 0.13 | 23.338 | 23.823 | 0.008 | 254.0 |
| mu[6] | 0.445 | 0.56 | 0.001 | 1.423 | 0.051 | 89.0 |
| bw[0] | 0.417 | 0.697 | 0.002 | 1.22 | 0.08 | 179.0 |
| bw[1] | 44.173 | 34.247 | 0.012 | 80.13 | 30.245 | 2.0 |
| bw[2] | 1.262 | 1.578 | 0.008 | 4.241 | 0.346 | 32.0 |
| bw[3] | 0.864 | 1.164 | 0.002 | 3.511 | 0.218 | 35.0 |
| bw[4] | 1.241 | 1.535 | 0.004 | 4.415 | 0.426 | 19.0 |
| bw[5] | 0.304 | 0.076 | 0.152 | 0.43 | 0.006 | 161.0 |
| bw[6] | 0.648 | 0.985 | 0.009 | 2.929 | 0.162 | 52.0 |
| w[0] | 1.831 | 8.71 | 0.001 | 4.979 | 0.424 | 231.0 |
| w[1] | 0.467 | 1.844 | 0.001 | 2.03 | 0.506 | 4.0 |
| w[2] | 1.365 | 6.084 | 0.001 | 4.867 | 0.379 | 41.0 |
| w[3] | 0.967 | 2.627 | 0.001 | 3.494 | 0.152 | 62.0 |
| w[4] | 0.901 | 2.221 | 0.001 | 3.711 | 0.116 | 23.0 |
| w[5] | 0.301 | 0.28 | 0.036 | 0.901 | 0.025 | 103.0 |
| w[6] | 1.111 | 2.54 | 0.002 | 4.1 | 0.134 | 82.0 |
| n | 0.029 | 0.025 | 0.002 | 0.061 | 0.023 | 1.0 |

**Sulphuric**

| Hyperparameter | mean | sd | hpd_2.5 | hpd_97.5 | mc_error | n_eff |
|---|---|---|---|---|---|---|
| mu[0] | 1.772 | 3.57 | 0.001 | 5.829 | 0.382 | 185.0 |
| mu[1] | 1.453 | 2.574 | 0.006 | 5.071 | 0.154 | 311.0 |
| mu[2] | 22.989 | 0.348 | 22.328 | 23.562 | 0.024 | 225.0 |
| mu[3] | 1.411 | 2.556 | 0.0 | 4.972 | 0.148 | 207.0 |
| mu[4] | 4.033 | 7.702 | 0.003 | 18.205 | 0.484 | 303.0 |
| mu[5] | 1.525 | 3.802 | 0.001 | 5.29 | 0.199 | 346.0 |
| mu[6] | 1.543 | 2.66 | 0.004 | 5.067 | 0.197 | 203.0 |
| bw[0] | 5.62 | 20.395 | 0.005 | 27.175 | 1.676 | 148.0 |
| bw[1] | 2.885 | 8.916 | 0.004 | 7.272 | 0.451 | 278.0 |
| bw[2] | 0.347 | 0.357 | 0.005 | 0.956 | 0.022 | 246.0 |
| bw[3] | 3.312 | 10.408 | 0.003 | 7.598 | 0.856 | 180.0 |
| bw[4] | 47.825 | 8.657 | 31.747 | 60.9 | 0.717 | 148.0 |
| bw[5] | 3.491 | 11.531 | 0.005 | 7.831 | 0.714 | 154.0 |
| bw[6] | 2.584 | 6.788 | 0.001 | 6.768 | 0.558 | 224.0 |
| w[0] | 0.482 | 0.956 | 0.001 | 1.774 | 0.048 | 200.0 |
| w[1] | 0.463 | 0.753 | 0.001 | 1.808 | 0.041 | 208.0 |
| w[2] | 0.245 | 0.427 | 0.007 | 0.978 | 0.031 | 215.0 |
| w[3] | 0.473 | 0.847 | 0.002 | 1.484 | 0.042 | 293.0 |
| w[4] | 0.14 | 0.033 | 0.082 | 0.206 | 0.003 | 147.0 |
| w[5] | 0.462 | 0.808 | 0.002 | 1.587 | 0.041 | 286.0 |
| w[6] | 0.583 | 1.186 | 0.003 | 1.912 | 0.099 | 195.0 |
| n | 0.154 | 0.067 | 0.026 | 0.247 | 0.006 | 120.0 |

**Unemployment**

| Hyperparameter | mean | sd | hpd_2.5 | hpd_97.5 | mc_error | n_eff |
|---|---|---|---|---|---|---|
| mu[0] | 58.239 | 12.011 | 60.011 | 61.384 | 3.346 | 90.0 |
| mu[1] | 0.705 | 0.34 | 0.154 | 1.395 | 0.173 | 3.0 |
| mu[2] | 1.142 | 1.295 | 0.168 | 4.885 | 0.391 | 14.0 |
| mu[3] | 0.755 | 0.913 | 0.292 | 1.064 | 0.143 | 12.0 |
| mu[4] | 0.227 | 0.433 | 0.051 | 0.554 | 0.151 | 1.0 |
| mu[5] | 40.609 | 0.062 | 40.482 | 40.71 | 0.031 | 5.0 |
| mu[6] | 20.163 | 0.193 | 19.775 | 20.467 | 0.016 | 145.0 |
| bw[0] | 0.408 | 0.643 | 0.129 | 0.673 | 0.071 | 11.0 |
| bw[1] | 26.706 | 33.094 | 0.035 | 128.937 | 17.066 | 3.0 |
| bw[2] | 5.543 | 2.269 | 1.766 | 9.23 | 0.881 | 7.0 |
| bw[3] | 0.321 | 0.675 | 0.017 | 0.458 | 0.156 | 6.0 |
| bw[4] | 2.36 | 1.738 | 0.024 | 5.186 | 1.003 | 3.0 |
| bw[5] | 0.039 | 0.031 | 0.014 | 0.085 | 0.008 | 8.0 |
| bw[6] | 0.444 | 0.15 | 0.224 | 0.691 | 0.101 | 3.0 |
| w[0] | 0.147 | 0.157 | 0.018 | 0.516 | 0.077 | 3.0 |
| w[1] | 0.009 | 0.067 | 0.0 | 0.004 | 0.011 | 7.0 |
| w[2] | 0.14 | 0.078 | 0.048 | 0.265 | 0.035 | 3.0 |
| w[3] | 0.659 | 0.267 | 0.226 | 1.158 | 0.074 | 14.0 |
| w[4] | 0.046 | 0.054 | 0.009 | 0.098 | 0.014 | 11.0 |
| w[5] | 0.697 | 0.424 | 0.274 | 1.544 | 0.147 | 9.0 |
| w[6] | 0.142 | 0.103 | 0.076 | 0.452 | 0.028 | 9.0 |
| n | 0.332 | 0.023 | 0.311 | 0.373 | 0.015 | 3.0 |

**Wages**

| Hyperparameter | mean | sd | hpd_2.5 | hpd_97.5 | mc_error | n_eff |
|---|---|---|---|---|---|---|
| mu[0] | 1.542 | 4.166 | 0.005 | 5.093 | 0.421 | 252.0 |
| mu[1] | 1.22 | 1.94 | 0.003 | 4.967 | 0.142 | 144.0 |
| mu[2] | 3.257 | 7.751 | 0.001 | 12.743 | 0.503 | 167.0 |
| mu[3] | 1.156 | 2.102 | 0.003 | 4.076 | 0.13 | 317.0 |
| mu[4] | 1.247 | 2.158 | 0.002 | 4.449 | 0.145 | 304.0 |
| mu[5] | 1.019 | 2.277 | 0.006 | 2.733 | 0.127 | 165.0 |
| mu[6] | 1.333 | 4.205 | 0.002 | 3.371 | 0.231 | 367.0 |
| bw[0] | 18.312 | 37.535 | 0.007 | 63.801 | 20.519 | 3.0 |
| bw[1] | 6.334 | 16.083 | 0.002 | 53.011 | 2.865 | 60.0 |
| bw[2] | 37.885 | 35.134 | 0.001 | 81.245 | 11.687 | 12.0 |
| bw[3] | 4.097 | 12.883 | 0.002 | 13.988 | 1.195 | 168.0 |
| bw[4] | 6.489 | 16.944 | 0.001 | 56.899 | 4.051 | 42.0 |
| bw[5] | 2.811 | 8.865 | 0.005 | 8.177 | 0.961 | 153.0 |
| bw[6] | 8.981 | 28.072 | 0.001 | 60.48 | 4.561 | 74.0 |
| w[0] | 0.523 | 1.176 | 0.004 | 2.09 | 0.173 | 142.0 |
| w[1] | 0.635 | 1.554 | 0.0 | 2.563 | 0.101 | 106.0 |
| w[2] | 0.433 | 1.163 | 0.004 | 1.467 | 0.086 | 172.0 |
| w[3] | 1.19 | 9.678 | 0.001 | 1.985 | 0.437 | 241.0 |
| w[4] | 0.565 | 1.152 | 0.004 | 2.121 | 0.06 | 230.0 |
| w[5] | 0.67 | 1.739 | 0.002 | 2.758 | 0.09 | 262.0 |
| w[6] | 0.967 | 4.25 | 0.001 | 2.997 | 0.205 | 261.0 |
| n | 0.15 | 0.032 | 0.083 | 0.197 | 0.005 | 72.0 |