# OpenReview forum: "Marginalised Gaussian Processes with Nested Sampling"
_NeurIPS.cc/2021/Conference — NeurIPS 2021 Poster_

### Official Review · Reviewer_whBU · 2021-07-12

**Rating:** 6
**Confidence:** 5

**Summary:**

The paper proposes using nested sampling to have a fully Bayesian GP. The idea is illustrated with spectral mixture kernel, and supported (to some extend) with experiments.

**Limitations And Societal Impact:**

See comment 6 above. The paper should say that while they strongly advice to use NS for SMK, the conclusion for simpler kernels like RBF/squared-exponential with limited number of hyperparameters required further study.

**Main Review:**

This is a paper to bring the technical of nested sampling that is well-used in physics and cosmology to the machine learning GP community. The paper is not very original and significant, given that nested sampling has been introduced previously in N(eur)IPS [Murray, 2005], and the authors of that paper have very deep knowledge of GP and are experts in this area. It is also widely accepted that margalised (or Bayesian) GP is *the correct* thing to to, though it can be somewhat challenging to execute in practice.

Nevertheless, this is a worthwhile paper, (re-)illustrating the significance of  Bayesian GP, esp in the context of the recently introduced spectral mixture kernel (SKM) which has multiple modes as illustrated by the authors. Also useful is to bring to attention the DYNESTY package. If the authors were to release their code, I can see that the code will be used frequently.

Section 4 is a particularly helpful advice for anyone attempting to use SKM. The paper is also well-written.

# Detailed comments
1. I prefer the term "Bayesian Gaussian Processes" rather than "Marginalised Gaussian Processes" in the title, but this is a matter of personal preference.
2. Given that the study is limited to SKM, this should be in the title.
3. Reorganised the paper to put SMK material in section 2 together with section 4.
4. It is paramount to cite and discuss [Murray, 2005].
5. I beg to differ from the sentence from line 232  to 233. The pattern for NS is too clear on along the axis in Figure 4, and then it blurs out along the edges. This is not what the generating function gives. It is nevertheless reassuring that all these are better than NKN.
6. Lines 262 to 263 tries to give something negative a positive light. What I read is that for simpler kernels/covariance-functions like the RBF/squared-exponential that are typically used in practice, there is no requirement for NS.
7. Lines  284 to 286 requires a correct heading.

Iain Murray, David MacKay, Zoubin Ghahramani, John Skilling. Nested sampling for Potts models. Part of Advances in Neural Information Processing Systems 18 (NIPS 2005)

Changes after author response
=======================
Removed the comment on NLPD from point 6.

**Time Spent Reviewing:**

4

---

> ### Author Response · Authors · 2021-08-10
> **Response to Reviewer whBU**
>
> Thank you for the detailed comments, we try and address each of them below:
>
> 1) Thank you for the suggestion - it is certainly a valid point and more well-known terminology.
>
> 2) While the focus is on the Spectral Mixture kernel mainly to demonstrate the benefits of multimodal marginalisation we also include results for the RBF kernel under both sampling schemes in the time-series experiments, hence we chose to use "with Nested Sampling" in the title. We believe Nested Sampling could prove beneficial for other complex kernels such as those involving changepoints.
>
> 3) Agreed.
>
> 4) We are certainly happy to include a mention of Murray's 2005 work on applying nested sampling to Potts models.
>
> 5) Perhaps there has been a slight misunderstanding regarding the performance of ML-II in Figure 4. Here the sampling methods aren't just "competitive with ML-II", they significantly outperform it. Further, the plot only visualises the mean of the posterior predictive distribution (which is computed by averaging over several means for the sampling methods) rather than the posterior predictive samples. The *blurriness* indicates uncertainty captured by the posterior predictive samples in regions with sparse training data. In either case, the sampling methods yield a much more reasonable extrapolation compared to canonical ML-II which appears to introduce artefacts.
>
> 6) For the particular tasks at hand, there is indeed little benefit to marginalisation of the simpler kernel - we will try to make this limitation more clear in the text. For more complex higher dimensional tasks, we anticipate there may still be considerable benefits to be found.
>
> 7) We agree that including a heading would be beneficial here.
>
> Thank you to the pointer to the Murray et al [2005] reference which will be included and discussed in related work.  We shall also clarify that the application to simpler kernels requires further study.

---

> > ### Comment · Reviewer_whBU · 2021-08-27
> > **Two-dimensional Pattern Extrapolation**
> >
> > Thank you for your reply. I realise that I misread the 216 of ML-II as 2.16, and will update the review accordingly.
> >
> > Now that it has been pointed out, I can see that the HMC is better than ML-II because of the sparse training data. However, the SM kernel is suppose to overcome this and extrapolate with the periodicities involved. Hence it is still rather disappointing in my view.
> >
> > For NS, I still think NS it is not doing a good job here. Given the number of training points in the central region, it overpredicts the values in the central region (i.e., the damping from $\sqrt{|x_1x_2|} is diminished) to around 3 and 4 rather the 0 and 1 of the ground truth. However, in contrast to HMC, it can infer the correct periodicities.
> >
> > It could be I still do not understand the plot in Figure 4. In which case, perhaps some other manner of illustrating the results could be helpful.

---

> > > ### Author Response · Authors · 2021-09-02
> > > **2d pattern extrapolation**
> > >
> > > Thank you for your message, we agree that the reconstruction has imperfections, and with more training data we expect the reconstruction would be much sharper - we will update the text to explain this in greater detail. We mainly wish to demonstrate that in situations of high epistemic uncertainty, the fully Bayesian formulation gives a more robust solution.

---

### Official Review · Reviewer_q4es · 2021-07-12

**Rating:** 7
**Confidence:** 4

**Summary:**

The paper addresses, the problem of simultaneously evaluating the marginal likelihood for Bayesian model selection and the inference of the posterior of hyperparameters in Gaussian process (GP) regression. To this end the authors utilize the nested sampling (NS). They show empirically, that when little data is present or the dimensionality of the problem is high NS outperforms the more common methods, such as marginal-likelihood (ML-II) maximization and Hamiltonian Monte-Carlo (HMC) sampling.

**Limitations And Societal Impact:**

Minors:
+ I would appreciate if runtimes could be reported for the experiments.
+ Notation in Algorithm1:
++ N -> n_L in the exponent
++ I think, the index i is a bit overloaded in meaning (running index in the initial set of live points, then a counter of samplings steps). Maybe change i to j in the loop and make clear, that it is a counter.
+ I’d prefer to write the convergence criterion additionally as a formula, because I needed to read the sentence in l.140 several times, and was still struggling.
+ L.79 completE
+ I found the paragraph from l.137 onwards not helpful. Especially, when it came to the rslice option. It would be nice, if the authors could add some more details, at least in the supplement.



**Main Review:**

The problem the authors consider is an important one in the Bayesian community. Apart from some minors (see below), I found the paper also well written. The experiments seem rigorously designed.

While I find the paper interesting and the experiments insightful, my main concern is the novelty of the proposed method. While it is true, that as the authors state NS isn’t well known in the ML community, it has been discussed to evaluate the marginal likelihood integral (see e.g. the thesis of Ian Murray [https://homepages.inf.ed.ac.uk/imurray2/pub/07thesis/murray_thesis_2007.pdf], which btw contains some nice illustrations to explain NS intuitively) and was also used in several works (of whose to be fair the manuscript is citing several). The main contribution of the current work comes from more rigorously comparing NS to the dominant solutions (ML-II, HMC), to solve the integral of interest, and using NS for  complicated priors (priors with spectral mixture kernels) in the GP regression setting.

Overall, I think the paper is and edge case, where on the pro-side stands a rigour empirical study to showcase the advantages of NS as a widely used model in the community, and on the contra limited novelty. In my opinion, the former slightly dominates, hence my rating.

## Update

After reading the authors response and the other reviews, I think, I was potentially underestimating the novelty of the work slightly. Hence, I will increase my score to 7.


**Time Spent Reviewing:**

5h

---

> ### Author Response · Authors · 2021-08-10
> **Response to Reviewer q4es**
>
> Thank you for the pointer to Iain Murray's PhD thesis, which indeed contains an elegant review of the nested sampling algorithm. We still believe there to be considerable novelty in our work, in part due to the application of nested sampling to Gaussian Processes. We will also emphasise that the nested sampling algorithm has progressed dramatically over the past decade. Prior to the introduction of multinest (Feroz et al 2009), it was not feasible to perform nested sampling on higher dimensional multimodal spaces. We believe it is valuable to highlight these advancements to the ML community.
>
> We report individual run-times per experiment (and averaged) for both methods HMC and Nested Sampling in the time-series experiment in Figure 6.
>
> We agree it would be clearer to modify the paragraph starting on L137, and to include a more detailed explanation in the Appendix and clarification of the convergence criteria. For reference, the 'rslice' option is one of six sampling methods available in the dynesty package (the full set of options can be found at https://dynesty.readthedocs.io/en/latest/quickstart.html#sampling-options).
>
> We will also clarify the notation in Algorithm 1 and fix the typing errors.

---

### Official Review · Reviewer_1r2q · 2021-07-13

**Rating:** 7
**Confidence:** 3

**Summary:**

The paper proposes nesting sampling techique for sampling from GP hyperparameter posteriors. Specifically, obtaining hyperparameters of the spectral mixture kernel is  discussed. Empirical results show that nesting sampling benefits GP performance while being computationally cheaper than HMC.

**Limitations And Societal Impact:**

The authors are aware of the limitations of their work and possible societal impact.


**Main Review:**

The paper addresses a problem of marginalizing GP hyperparameters. The proposed technique is nested sampling of the hyperparameters. Nested sampling (NS henceforth) was developed previuosly, however the application of the technique to GPs is original work. The authors thoroughly cite related work.

The work claims that NS improves GP behavior compared to a point estimate of the hyperparameters. On the other hand, NS takes significantly less computation time than exploring the posterior with HMC. Performance comparison between NS and HMC seems to depend largely on the kernel and the dataset. There is also a comparison with Neural Kernel Network.
The claims are supported by experiments.

The paper is very well-written and organized. The supplementary material provides settings to reproduce the results. Here are a some issues:
p. 3 l. 79: complet -- typo
It could also benefit to have larger pictures or larger fonts in the pictures, because they have to be ridiculously zoomed-in to be seen.
Also, please provide a clearer explanation of the choice of the uniform prior for frequencies larger than the fundamental.

The results are important because they provide a relatively low-cost (compared to HMC) technique to marginalize SM kernel hyperparameters which improves the performance of the GP. The SM kernel is indeed a very expressive family.

Overall, very good paper.

**Time Spent Reviewing:**

4

---

> ### Author Response · Authors · 2021-08-10
> **Response to Reviewer 1r2q**
>
> We are grateful for the positive feedback, we are pleased the manuscript was found to be valuable and well written. We will certainly address the highlighted issues regarding the typing errors and font sizes for captions.
>
> As requested, we will elaborate on the justification for a uniform prior at frequencies above the fundamental. To very briefly summarise here - this stems from the effect of aliasing, where contributions from frequencies above the Nyquist frequency are projected down to below the Nyquist frequency. Hence, if the mean frequency has a broad prior $\mu \sim LogNormal[a, b]$ -> this ends up getting reduced to $\mu  \pmod{\nu_N} $,  so the implied prior is approximately equivalent to a Uniform upper bounded at the Nyquist frequency.

---

> > ### Comment · Reviewer_1r2q · 2021-08-31
> > **1r2q  response to authors**
> >
> > I thank the authors for taking time to address my and other reviewers' review. After reading the discussion, I still recommend acceptance, my score remains 7.

---

### Official Review · Reviewer_XcBt · 2021-07-15

**Rating:** 7
**Confidence:** 4

**Summary:**

The paper suggests using the nested sampling algorithm to marginalize hyperparameters of Gaussian processes with spectral mixture kernels. The motivation is that the posterior over hyperparameters of such kernels is highly multimodal and nested sampling is known to work well with multimodal distributions. Furthermore, practical priors for the parameters of spectral mixture kernels are suggested. The proposed nested sampling driven marginalised Gaussian process is compared to NUTS driven marginalised Gaussian process and the Gaussian process with hyperparameters estimated by maximum likelihood on a set of low-dimensional benchmark problems. The comparison is in terms of negative log predictive density. It is shown that the proposed method usually outperforms other methods while also performing faster than NUTS.

**Limitations And Societal Impact:**

The limitations and societal impact are adequately addressed.

**Main Review:**

The paper highlights the method of nested sampling, popular in physics, as a method for marginalizing over hyperparameters of spectral mixture kernels. This is a valuable contribution. The conducted empirical study is convincing in showing that the method is indeed promising in this new setting. The method is only benchmarked in low dimensions but may be useful in the time series modeling problems which are quite broad and important themselves.

# Originality
As far as I can tell, the contribution is novel.

# Quality
The claims are supported by an extensive empirical evaluation. On the downside, the implementation is not provided and said to be proprietary.

# Clarity
The paper is very well written. The most serious problem of the sort spotted by me in the paper is fonts being too small in some figures (especially in Figure 2).

# Significance
The method suggested by the paper and the corresponding empirical study will be valuable for machine learning practitioners who use low-dimensional Gaussian processes. Mostly, this work is relevant for time series modeling. Regrettably, this positive impact is somewhat limited by the fact that the code will not be openly released for people to be able to quickly and easily apply the method to their problems.

# Remarks
- Line 17. I suggest giving a reference or briefly discussing the abbreviation ML-II, it might not be a universally known one.
- Line 39. “lies not its”->”lies not in its”.
- Line 44. “some the”-> “some”.
- Lines 67-68. Phrase “the final form of the posterior predictive in a marginalised GOs is a mixture of Gaussians” is not rigorous because it is only approximately so, I suggest reflecting this in text.
- Line 75. “consider”->“considered”.
- Line 76. “demonstrate”->”demonstrated”.
- Line 79. “complet”->”complex”.
- All figures except Figure 3 need larger fonts.


**Time Spent Reviewing:**

7

---

> ### Author Response · Authors · 2021-08-10
> **Response to Reviewer XcBt**
>
> We are pleased that the proposed approach was found to be promising, and that the empirical evaluations were valuable. We will certainly incorporate the amendments regarding the text and caption fonts, along with fixing the typos mentioned.
>
> **On release of the code**: We are happy to release code for the models along with the supplementary material. The proprietary aspect of the code relates only to the benchmarking procedure.
>
> **On low-dimensional benchmarks**: We indeed chose to demonstrate marginalisation on time series (1d) and a pattern task (2d) settings as it is easier to interpret priors over hyperparameters in these low dimensions, as well as gaining insight through exploring the marginal likelihood surface. Tackling higher dimensional tasks would serve as an interesting topic for future exploration.

---

### Author Response · Authors · 2021-08-10
**General response to all reviewers**

We would like to thank all reviewers for their time and positive feedback on the manuscript. Here we include a short paragraph which clarifies the overall gist and significance of the paper.

A novel application of the Nested sampling (Skilling 2004) technique to approximate the hyperparameter posterior in Gaussian processes. The main reason Nested sampling is an attractive choice is because it enables gradient free multi-modal marginalisation and the predictive posterior averages over many complimentary solutions in function space. Multi-modal marginalisation is challenging for gradient based samplers like HMC and its variants like NUTS, although by running several chains one can overcome this constraint and access multiple modes. In practice, it is more convenient to use Nested sampling or similar techniques which don't jump between modes but zone into them by design. It also brings into purview of the ML community a compelling method which can be applied and tested in several different paradigms like Bayesian neural nets, deep GPs and other extended models.

---

### Decision · Program_Chairs · 2021-09-27

**Decision:**

Accept (Poster)

**Comment:**

The paper proposes the use of nested sampling (NS) for inference in Gaussian process (GP) models with Gaussian likelihoods and shows the benefits of this fully Bayesian approach over competing methods such as HMC and type-II marginal likelihood hyper-parameter estimation. Although NS has been studied previously and the evaluation of the proposed method is mainly focused on the spectral mixture kernel and low-dimensional settings, all the reviewers recommend acceptance and I agree, as the paper further strengthens the machinery to carry out inference in GP models. I note and appreciate the authors’ commitment to release the corresponding code.